# Pre-Clinical Models to Study Human Prostate Cancer

**DOI:** 10.3390/cancers15174212

**Published:** 2023-08-22

**Authors:** Martin K. Thomsen, Morten Busk

**Affiliations:** 1Department of Biomedicine, Aarhus University, 8000 Aarhus, Denmark; 2Department of Experimental Clinical Oncology, Aarhus University Hospital, 8200 Aarhus, Denmark; morten@oncology.au.dk; 3Danish Centre for Particle Therapy, Aarhus University Hospital, 8200 Aarhus, Denmark

**Keywords:** mice, prostate, androgen, prostatic intraepithelial neoplasia, prostatic neoplasms, clustered regularly interspaced short palindromic repeats, adenocarcinoma, transgenes, tumor suppressor, allografts

## Abstract

**Simple Summary:**

The incidence of prostate cancer is rising, primarily due to its prevalence among elderly men, and with increased life expectancy, these numbers are expected to continue increasing. Prostate cancer can remain indolent for many years, but treatment options are limited once the cancer progresses to an aggressive stage. Pre-clinical models have played a vital role in gaining insights into this particular cancer, revealing essential information about the molecular alterations that drive cancer progression. The mouse model has been invaluable for studying prostate cancer, employing both genetically modified strains and the inoculation of prostate cancer cells. This review will focus on the development of pre-clinical models for studying prostate cancer and discuss future directions for enhancing our understanding and developing interventions for prostate cancer.

**Abstract:**

Prostate cancer is a common cancer among men and typically progresses slowly for several decades before becoming aggressive and spreading to other organs, leaving few treatment options. While large animals have been studied, the dog’s prostate is anatomically similar to humans and has been used to study spontaneous prostate cancer. However, most research currently focuses on the mouse as a model organism due to the ability to genetically modify their prostatic tissues for molecular analysis. One milestone in this research was the identification of the prostate-specific promoter Probasin, which allowed for the prostate-specific expression of transgenes. This has led to the generation of mice with aggressive prostatic tumors through overexpression of the SV40 oncogene. The Probasin promoter is also used to drive Cre expression and has allowed researchers to generate prostate-specific loss-of-function studies. Another landmark moment in the process of modeling prostate cancer in mice was the orthoptic delivery of viral particles. This technology allows the selective overexpression of oncogenes from lentivirus or the use of CRISPR to generate complex loss-of-function studies. These genetically modified models are complemented by classical xenografts of human prostate tumor cells in immune-deficient mice. Overall, pre-clinical models have provided a portfolio of model systems to study and address complex mechanisms in prostate cancer for improved treatment options. This review will focus on the advances in each technique.

## 1. Pre-Clinical Models for Human Prostate Cancer

The prostate is a sex-specific organ that shows significant differences in morphology and secreted proteins among mammals. The secreted proteins have specific functions that can vary between species, and in humans, only PSA (Prostate-Specific Antigen) and PSMA (Prostate-Specific Membrane Antigen) are expressed, serving as biomarkers for prostate inflammation and prostate cancer [1]. The human prostate has a ball-shaped morphology with distinct zones [2]. Dogs and pigs also exhibit a similar morphology, while rodents have four well-defined bilateral lobes [3,4]. Among these species, only dogs develop sporadic prostate cancer but typically at a late age, which makes it challenging to conduct animal studies on prostate cancer (PCa) [5]. The choice of an appropriate pre-clinical model for PCa has been a topic of extensive discussion. Most of the research in this field has been carried out in mice, which can be genetically modified to study PCa. Particularly, the dorsal and lateral lobes of the mouse prostate resemble the peripheral zone of the human prostate, where most cancers arise [6]. Although no perfect pre-clinical model exists, the mouse is preferred due to its amenability to genetic modifications, allowing for the study of cancer initiation and progression within a relatively short time frame.

## 2. Evolution of Mouse Models for Prostate Cancer

### 2.1. Prostate-Specific Promoters

Genetic mouse models have transformed cancer studies, but in the case of prostate cancer, which progresses slowly, mice with broad genetic alterations often develop cancer in organs other than the prostate [7,8]. Therefore, to study prostate cancer, tissue-specific gene alterations must be applied, and researchers have focused on finding prostate-specific promoters to drive transgene expression. In the 1990s, Greenberg and Zhang utilized the promoter from the rat secreted protein Probasin to drive tissue-specific expression in the mouse prostate [9]. Through delineation of the promoter, they identified a fragment, ARR2PB, that exhibited high specificity and strong expression when four elements were used due to presence of a binding motif for the androgen receptor (AR) [10]. Using this modified promoter, they generated a Cre-expressing mouse line called PB4Cre, which is the most commonly used Cre line for studying prostate cancer [11]. However, the expression is not uniform across the prostatic tissues; highest expression is seen in the dorsal and lateral lobes, but only 10–20 percent of cells in the anterior and ventral lobes show expression [11]. Other prostate-specific Cre-expressing lines have been generated, including PSA-Cre and Nkx3.1, with both genes driven mainly by AR expression [12,13,14]. The Nkx3.1-Cre line has further been developed as a tamoxifen-inducible Cre line, enabling researchers to control the timing of Cre activation or to perform a labeling experiment with a pulse of Cre activation by tamoxifen treatment [15]. These diverse prostate-specific Cre-expressing mouse lines are indispensable for genetic studies of prostate cancer and are commonly employed.

### 2.2. Prostate-Specific Gene Alteration

The development of a prostate-specific promoter allowed researchers to generate mice with oncogene expression restricted to the mouse prostatic epithelium. One of the first mice established using this technique was the TRAMP mouse, in which the SV40 large and small antigens were expressed from the Probasin-modified promoter. This generated a model of prostate cancer that has been widely used. The TRAMP mouse model exhibits rapid tumor progression with high-grade PIN at 12 weeks and displays metastasis, reaching a humane endpoint at around 30 weeks of age [16]. This model has enabled researchers to study PCa progression and to conduct pre-clinical studies to interfere with cancer progression by targeting androgen and other pathways [17]. However, it should be noted that the TRAMP model displays heterogenic cancer, with aggressiveness driven by neuroendocrine cells rather than the transformed prostatic epithelium [18]. This limits the use of the TRAMP mouse, as only a minor part of human PCa consists of neuroendocrine-driven cancer, even though this cancer form is on the rise [19].

Only a few genetically modified mice exhibit alterations in the prostatic epithelium [20]. Loss of Nkx3.1 results in prostatic intraepithelial neoplasia (PIN) development, but not until around 1 year of age, and this condition does not progress to cancer (Figure 1) [21]. Additionally, the heterozygous loss of Pten induces PIN formation at 6 months of age [22]. Since both Nkx3.1 and Pten are commonly mutated in human PCa, studying these gene alterations in the mouse prostate is biologically relevant as a model for human PCa (Figure 1) [23]. The group led by Hong Wu generated a mouse with a conditional allele for Pten and crossed it with the PB4Cre line to achieve prostate-specific depletion of Pten. This resulted in a robust mouse model for PCa with early PIN formation and progression to cancer, eventually leading to metastatic formation at a late stage [24]. The background of the mouse strain interfered with the onset of cancer, but most studies are now conducted in C57BL/6 mice, which exhibit a delayed phenotype [25]. Pten is a phosphatase that dephosphorylates PIP3, thereby antagonizing PI3K. Loss of Pten increases phosphorylation through the PI3K pathway, activating AKT and mTOR, which in turn leads to increased proliferation of the prostatic epithelium—a hallmark of cancer initiation [26]. Therefore, loss of Pten or activation of its downstream pathway is considered an essential event in prostate cancer and is applied to most mouse models of PCa to ensure transformation and proliferation of the prostatic epithelium (Figure 1).

### 2.3. Targeting Multiple Genes in the Mouse Prostate

The use of conditional alleles in combination with prostate-specific Cre lines has allowed researchers to explore specific gene functions in the prostate. However, depletion of many different genes has shown either no or very minor phenotypical changes in the prostate, which often occur at a late stage [27,28,29,30]. This has led the scientific community to interbreed the strain containing a conditional allele for Pten with the conditional strain of interest. One of the initial studies was conducted by Pandolfi’s group, where the combination of Pten and Trp53 loss was investigated. These mice exhibited an accelerated phenotype, revealing a clear positive genetic alteration associated with the loss of these two tumor suppressor genes. The mice reached a humane endpoint at around 6 months of age, characterized by a significantly enlarged prostate [31]. However, the humane endpoint is not directly related to cancer but rather to the obstruction of kidney function [31]. Cancer progression can be further accelerated in this model by the additional deletion of Rb1, whereas the loss of either Trp53 or Rb1 alone results in an indolent phenotype at a later age (Figure 1) [32]. Mutations in TP53, PTEN, and RB1 are commonly found in human PCa, and thus the mouse models reflect human PCa biology [33,34].

Another interesting mouse study was conducted by DePhino’s group, where the conditional allele for Smad4 was combined with Pten loss [35]. These mice exhibited rapid tumor progression with metastatic formation in various organs, including the bones [36]. Metastasis, and in particular, bone metastasis, are rare events in pre-clinical models for PCa, presumably due to the presence of a large primary tumor, which defines the humane endpoint. However, metastases are important features of PCa and are highly relevant for studying PCa progression, as patients with metastatic PCa have a poor prognosis (SEER Cancer). Other genes have been studied in combination with Pten loss, contributing to the understanding of how gene alterations in PCa can accelerate the development or differentiation of mutated cells [27,28,29,30,37,38,39,40,41,42]. A commonly mutated gene in human PCa is KMT2C, but mice with the loss of Kmt2c alone do not display a clear cancer-related phenotype. However, when combined with Pten loss, metastatic formation is often observed [43], and we have unpublished data that confirm the role of Kmt2c in the dissemination of prostate cancer. Overall, the application of mouse models with the loss of Pten has generated essential knowledge about PCa and gene alterations, which do not manifest phenotypically when the genes are deficient individually (Figure 1).

### 2.4. Gain-of-Function Studies for PCa

Pre-clinical mouse models with gain-of-function mutations have been generated in a similar fashion to the aforementioned TRAMP mouse. This includes a mouse model with androgen expressed under the Probasin promoter, where a point mutation has been introduced to the reading frame. This mutation is often found in human PCa with androgen-insensitive cancer. These mice developed aggressive cancer with the formation of metastasis after 50 weeks, confirming the oncogenic properties of mutated AR [44]. Genomic translocation is a common event in PCa, particularly the fusion between the promoter of the AR-regulated TMPRSS2 and ETV1 or EGF genes [45]. Multiple mouse models have been generated that overexpress either ETV1 or EGF under the modified Probasin promoter, resulting in PIN formation and possible invasive adenocarcinoma at late stages [46,47,48].

Amplification of MYC is observed in numerous of human cancers and is a frequent occurrence in prostate cancer (PCa), emphasizing its oncogenic function in this particular cancer [49]. MYC’s role in PCa has been modeled using gain-of-function models with prostate-specific expression. Overexpression of MYC alone is sufficient to transform the prostatic epithelium, leading to the formation of PIN and adenocarcinoma [50,51]. However, when combined with the loss of Pten or of both Pten and Trp53, MYC overexpression accelerates cancer progression and facilitates cancer dissemination [52,53]. These findings demonstrate that MYC amplification in human cancer is mirrored in mouse models of PCa, and RNAseq data from various mouse models of PCa confirm the upregulation of MYC target genes [54].

In humans, point mutations in the E3 ligase SPOP are seen in around 10% of PCa cases and are an early event. A mouse model of a SPOP point mutation has been generated by two independent groups by the expression of SPOPF133V from the Rosa26 locus controlled by lox–stop–lox. This mouse line has been interbred with the Probasin Cre line to generate prostate-specific expression. A minor phenotype was observed, but by intercrossing with the conditional allele for Pten, invasive adenocarcinoma developed [55,56]. SPOPF133V increases the expression of AR-regulated genes and underpins the oncogenic property in PCa [55]. Other gain-of-function studies have been conducted in mice to understand the principles of amplified genes in PCa initiation and progression. However, most models require transgenic mice and tissue-specific expression to study the function in the prostatic epithelium, which can be time consuming.

## 3. Orthotopic-Manipulation-Based PCa Models

The mouse prostate differs from the human prostate by having four bilateral lobes. The anterior lobe contains a large lumen, which allows researchers to inject a small volume of approximately 25 µL for orthotopic delivery. This is performed through a small operation involving an insertion into the abdominal skin, followed by the lifting of the seminal vesicles to expose the anterior lobes of the prostate [57]. In a study by Leow et al., orthotopic delivery of adenovirus to the mouse prostate was employed to demonstrate tissue-specific delivery without the need for an additional transgenic mouse line such as Probasin Cre. The researchers conducted the experiment in a reporter mouse line to activate the expression of beta-galactosidase [58]. We applied a similar method to deliver adenovirus expressing Cre, inducing conditional loss of Junb and Pten in the prostatic tissues to avoid the use of a Cre-specific mouse line [28].

The group led by Trotman also utilized orthotopic delivery to the mouse prostate to establish a novel mouse model of prostate cancer called RapidCaP [52]. In their approach, they used lentivirus to express Cre protein and luciferase to track cells with an integrated virus. Lentivirus was applied to mice with double conditional alleles for Trp53 and Pten, allowing them to monitor tumor progression and cancer dissemination through the illumination from the luciferase. They demonstrated that tumor progression was driven by Myc activation, leading them to overexpress Myc through orthotopic lentivirus delivery to the mouse prostate as a proof of concept. This study highlighted the effectiveness of applying orthotopic lentivirus delivery for studying prostate cancer, as transgene expression can proceed in the transformed cells [42,52].

Inflammation in the prostate is recognized as a risk factor and a marker for PCa [59]. However, only a few pre-clinical models have addressed the implications of prostatic inflammation on PCa initiation and progression. Two studies have employed bacterial strains that were inoculated into the prostates of wildtype mice. Both studies have demonstrated alterations in the inflamed gland, including the decreased expression of Nkx3.1, increased proliferation, and the formation of fibrosis [60,61]. These studies suggest that inflammation can disrupt tissue homeostasis and potentially initiate PCa, but further research in this aspect is required.

### CRISPR-Generated PCa Models

CRISPR technology has been applied to generate pre-clinical models in different species. In mice, CRISPR is used to genetically modify embryonic stem cells and is also applied to induce somatic mutations to study cancer in specific organs [62]. For somatic mutations, Cas9 transgenic mice have been used to avoid the delivery of Cas9 and to ensure biosafety without the combined delivery of Cas9 and sgRNA [62]. To study prostate cancer (PCa), we applied CRISPR through the orthotopic delivery of adeno-associated virus (AAV) to the anterior mouse prostatic lobe [27,37]. The advantage of AAV is that it rarely integrates into the host cell’s genome and has very little immunogenicity [63]. However, the AAV genome is small and cannot exceed 5 kb, which imposes limitations on the genomic cargo that can be delivered. Therefore, we took advantage of Cas9 transgenic mice with either conditional expression of Cas9 and EGFP for visualization or Cas9 expression from a ubiquitin promoter [27,64,65]. Each line has advantages: with conditional Cas9-EGFP expression, only cells transduced with a virus expressing Cre will express Cas9 and EGFP. However, Cre expression will have to be delivered by the virus or, as an alternative, the conditional Cas9 mouse line needs to be interbred with a Cre line to obtain prostate-specific expression. The ubiquitin Cas9-expressing line does not have these limitations but lacks EGFP expression, and tumors can occur in other organs if the virus has spread outside the prostate [62]. 

One advantage of CRISPR-generated tumors is the ability of multiplex sgRNAs to target multiple genes simultaneously in the same prostatic epithelial cell. This mimics the occurrence of multiple mutations over time in the same cell, allowing for the study of genetic interactions. Using this method, it is feasible to mutate 3–4 genes simultaneously, and we have successfully targeted up to 8 genes (Cai et al., under revision), which would not be possible through the traditional interbreeding of conditional mouse strains. However, there are limitations associated with CRISPR-generated tumors. First, the AAV virus only allows for a cargo of 5 kb, which is equivalent to 10–12 sgRNAs or 6 sgRNAs and a Cre expression cassette. Second, CRISPR-induced mutations are stochastic and can result in the insertion or deletion of bases, often leading to truncated proteins. However, they can also generate functional proteins with a loss or gain of amino acids. Furthermore, the expression of sgRNAs does not always result in a double-stranded break, which can lead to tumors with different mutation profiles among the animals. Therefore, it is necessary to sequence the mutation sites for each tumor to validate the appropriate gene mutations before analyzing the phenotype [62]. In lung cancer studies, CRISPR technology has allowed the generation of tumors with combined loss- and gain-of-function mutations [65,66]. Future studies in the prostate can take advantage of similar methodologies to model different gain-of-function mutations commonly observed in human PCa in combination with the loss of tumor suppressor genes.

Orthotopic delivery of viral particles to induce PCa has advantages over traditional mouse models. Since the orthotopic delivery targets somatic cells, the virus can be titrated to transduce only a few cells. This allows for clonal expansion and the development of a defined tumor, whereas models with Cre expression in the majority of prostatic epithelium can result in hyperplastic lesions with limited progression to cancer before reaching the humane endpoint [57]. This has been observed in mice with double mutations of Pten and Trp53, where mice developed kidney problems due to a dysfunction in emptying the bladder as the entire prostate became significantly enlarged [31]. The most commonly used Cre line is the PBCre4, where expression occurs from 1–3 weeks after birth, allowing for genetic modifications in the pre-adult organ [11]. In contrast, orthotopic delivery is typically performed in adult mice, and researchers can mimic when in the life cycle the cancer is induced. Thus, orthotopic delivery offers an advantage by inducing cancer in a specific subset of cells, enabling clonal expansion while also being temporally aligned with the mouse’s age.

## 4. Pre-Clinical Models by Cell Implantation

### 4.1. Classical Cell Line-Derived Xenografts

The most widely used pre-clinical models for studying PCa are xenografts of human PCa cells. In this model, previously established patient-derived monoclonal cell lines are grown in immunocompromised mice, offering several advantages. Commercially available cell lines are well-characterized, allowing for a direct comparison between in vitro determined molecular vulnerabilities, druggability, and their translation into potential treatments in complementary in vivo scenarios. Such research is invaluable and has played a central role in cancer research for several decades, and the importance of these models should not be dismissed.

The most commonly used prostate cell lines are PC3, DU145, and LNCaP. However, the first two cell lines are not androgen-sensitive and do not secrete PSA. Typically, cells are inoculated subcutaneously (SC), often in the flank, which is a relatively simple procedure and enables easy caliper-based assessment of tumor growth (Figure 2, Table 1). This method is highly valuable for evaluating the efficacy of interventions. SC inoculations also generally result in higher take-rates compared with orthotopic inoculations since a larger number of cells can be administered. Additionally, SC tumors can grow to a considerable size without compromising animal welfare, which is crucial for certain applications such as the development of functional imaging in oncology (e.g., PET) [67]. However, SC tumors may develop differently than patient tumors. Studies have shown that orthotopic prostate tumors exhibit better perfusion and less hypoxia compared with SC growing tumors [68]. This difference in hypoxia levels is significant because hypoxia is associated with aggressive growth, metastasis formation, and reduced treatment sensitivity. Similar observations have been made in lung tumors [69], and it may be an inherent characteristic of growth in the SC niche. Nevertheless, in certain contexts, maintaining similarity with the average patient tumor is not crucial or even desirable. For example, when developing hypoxia-targeting treatments, tumors that mimic the extent of hypoxia in the most hypoxic patient tumors, rather than the typical patient, are optimal. As an example, the development and testing of the radiosensitizer Nimorazole, which is now standard therapy for head and neck cancer patients in some countries, was based exclusively on mouse studies using a single SC tumor model [70]. Additionally, for localized treatments such as radiotherapy or hyperthermia, an SC location in an extremity (foot/thigh) is extremely helpful to ensure accurate dosing and to minimize side effects on crucial organs, which may limit treatment intensity and the testing of potentially curative treatments [71]. Performing similar research in a relevant orthotopic model would pose significant challenges due to several factors: (1) limitations related to tumor expansion as per humane endpoint constraints; (2) the difficulty in administering potentially curative doses without causing unacceptable harm to surrounding sensitive tissue; (3) the impracticality of conducting accurate tumor response assessments without labor-intensive imaging procedures. Consequently, the model that best replicates the clinical disease both biologically and anatomically might not always be the most suitable for the initial testing and development of novel treatments, particularly during the early stages of investigation.

### 4.2. Patient-Derived Xenografts

Although classical xenograft tumor models are established from cells originating from patients, the term “Patient-derived xenograft” (PDX) specifically refers to models that are directly established from patients by inoculating tumor material (e.g., a biopsy or a surgical specimen) into mice, typically through subcutaneous injection. This bypasses the step of establishing cell lines (Figure 2, Table 1). The major advantage of PDX models is that they may more faithfully recapitulate the biology of the original tumors, including the cellular and microenvironmental heterogeneity. This, in turn, can provide valuable information on their biology, microenvironment, biomarkers, treatment sensitivity, and more [72].

PDX models have been successfully established from a variety of cancers, but the establishment of prostate PDX tumors has proven to be problematic compared with tumors from many other sites [73]. One major obstacle is the significantly lower levels of androgens in male mice compared to men. One study showed that testosterone supplementation increased the take-rate, but it still remained low (10%) compared with other cancers [74]. A low take-rate is problematic not only because it is labor-intensive and costly but also because more advanced prostate cancers tend to have a higher take-rate, leading to an inherent bias. It is important to consider this issue for all PDX models. Another concern with PDX models is the potential loss of crucial traits of the original patient tumor during passaging in mice (Table 1). This can be tumor-type specific, but in general, it is recommended to minimize passaging and to conduct a thorough histopathological examination at different passages [75]. As an alternative to PDX models, 3D in vitro models of patient-derived organoids can be generated, and this method is rapidly developing [76]. In vitro models offer advantages such as controlled drug delivery and molecular analysis, which can be performed under controlled conditions compared to in vivo settings. However, in vivo settings more accurately replicate the patient’s situation, which is advantageous for numerous studies.

### 4.3. Allograft and Metastatic Models for PCa

Until now, immunotherapeutic strategies against prostate cancers have largely failed [77], and further research is warranted. The application of xenograft models for immunotherapy research is clearly challenging when using human-derived cell lines, even though efforts are being made to humanize immunocompromised mice to establish a patient-like immune system [78]. Instead, mouse-derived tumor cells can be allografted to syngeneic immunocompetent mice to study immune surveillance of the tumor. However, there have not been many cell lines derived from mouse PCa, and this area generally requires further research. One cell line was derived from a transgenic Myc-driven PCa on a C57BL/6 background. This cell line showed clear expression of prostate epithelial markers and responded to castration. Furthermore, when injected intracardiacally, this cell line could form bone metastases, which is highly relevant when studying PCa metastasis [79]. The Abate-Shen group also established cell lines from primary and secondary PCa, which exhibited loss of Pten and a gain-of-function mutation of Kras. Their work revealed that an increased Myc gene expression signature promoted metastasis formation in the bone but also that inoculation of the cells by intracardiac injection favored metastasis in the bone (Figure 2, Table 1) [54].

Intracardiac injections have also been performed with human PCa cell lines in immunodeficient mice. Specifically, PC3 cells have been injected intracardiacally, resulting in the formation of bone metastasis much more efficiently than by orthotopic implantation [80,81]. Some researchers have taken a step further and performed femur grafts to ensure the establishment of tumors in the bone. This model is useful for studying PCa cells in the bone microenvironment but cannot be used to investigate the homing of prostatic cells to the bone (Figure 2) [81].

## 5. Future Perspectives

Pre-clinical models of PCa have proven crucial for elucidating the molecular mechanisms involved in PCa initiation and progression. Genetic mouse models and the interbreeding of different genetic modifications have shed light on key mutations and their implications for PCa. A tremendous effort has been made to generate prostate-specific genetic alterations, enabling researchers to ask specific questions focused on PCa. The genetic mouse models have been complemented by xenograft and patient-derived xenograft (PDX) models, which have brought human cancer into pre-clinical settings. Future work will be driven by the aim to replicate human PCa and to develop effective treatments. In vivo models are essential for this purpose, as drug delivery poses a significant challenge in treatment, and the study of immune therapy necessitates appropriate models. Therefore, more refined models will need to be developed, including the orthotopic delivery of viral particles to generate clonal tumor expansion, which is valuable for understanding cancer evolution. These genetic mouse models should also be complemented by the generation of cancer cell lines, particularly derived from the mouse prostate at different stages of cancer. These cell lines are important for allograft studies, with a focus on the tumor microenvironment and the testing of cancer interventions.

## 6. Conclusions

Pre-clinical models have significantly contributed to our understanding of the molecular alterations that initiate and drive the progression of PCa. This is especially crucial since PCa can remain indolent for many years before progressing to an aggressive cancer. The mouse has emerged as an essential model for grafting human cells and utilizing genetically modified strains. Mutations frequently identified in human PCa have been studied in this model, providing valuable insights into the molecular pathways and implications for cancer progression. In future PCa research, pre-clinical models will continue to play an essential role in the development of interventions targeting disease progression, with the elimination of cancer as the primary objective.

## Figures and Tables

**Figure 1 cancers-15-04212-f001:**
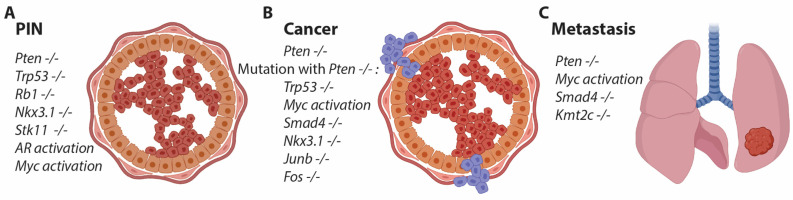
Genetically modified mice exhibit different phenotypes in the prostatic epithelium. (**A**) In certain cases, a single gene alteration can induce transformation of the epithelium and lead to the development of prostatic intraepithelial neoplasia (PIN) over time. (**B**) Loss of Pten is a hallmark in mouse models of prostate cancer (PCa), and this single gene alteration can drive PIN formation, eventually progressing to PCa and metastasis in secondary tissues. The combined loss of Pten with other gene mutations, including the loss of Trp53, Nkx3.1, members of the AP1 gene family, and others, has been shown to accelerate the phenotype. (**C**) Certain genetic alterations, when combined with the loss of Pten, promote the formation of metastasis.

**Figure 2 cancers-15-04212-f002:**
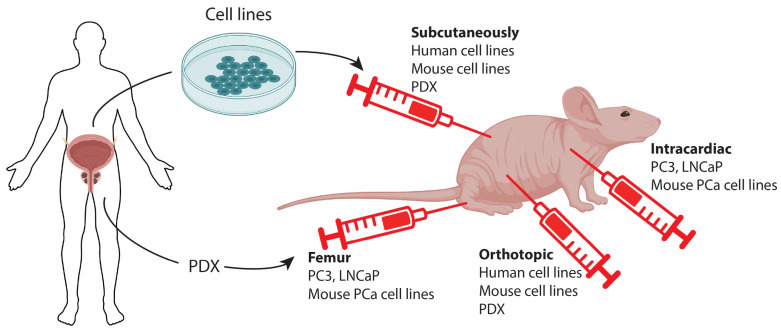
Isolated cells from human prostate cancer (PCa) can be inoculated into immunodeficient mice. Classical cell lines are derived from tumor tissues and grown in vitro before being grafted onto a mouse. Alternatively, patient-derived xenografts (PDX) can be used, where tumor tissues are directly used for grafting. Tumor cells can be inoculated into different locations, each with its own advantages. Subcutaneous grafting is commonly performed, allowing for the easy monitoring of tumor progression and growth to a large size with minimal implications for the animal. Orthotopic grafting involves grafting tumor cells into the mouse prostate, enabling the development of tumors in the organ of origin. Intracardiac or tail vein injections allow for the dissemination of tumor cells and the development of tumors in other organs, serving as a model for metastatic cancer. This can be further expanded by femur injection to establish metastasis in the bone of the recipient mouse.

**Table 1 cancers-15-04212-t001:** Advances and limitations with cell-derived in vivo models of PCa.

	Biological Inoculation Material/Induction Technology	Key Advantages	Key Limitations
Classical cell line-based xenograft models	Cell lines: LNCaP DU145 PC3	Patient-derived Well characterized Allows complementary in vitro experiments, which is valuable for drug development Tumor growth is easy to monitor when SC inoculation is used	Cells may have lost hallmark features of human disease Absence of a functional immune system, which may influence biology and treatment response Few models available
PDX models	Patient biopsy	Patient-derived Metabolic phenotypes are more reflective of human disease, since a lengthy in vitro selection process is avoided Patient-to-patient and intratumoral heterogeneity are better preserved Tumor volume is easy to monitor when SC inoculation is used	Expensive and time-consuming, not least for prostate PDX models where the take-rate is low compared with other tumors Heterogeneity and patient similarities may be lost through repetitive passaging and cryopreservation Absence of a functional immune system
Mouse derived cell lines	Cell lines: B6CaP NPK	Can be used to allograft Functional immune system, allowing immunotherapy	Few cell lines Has been cultured in vitro and can have lost features of PCa

## Data Availability

The data presented in this study are available in this article.

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
