# Peer review of "Pre-Clinical Models to Study Human Prostate Cancer"

_cancers, 2023, doi:10.3390/cancers15174212_

Round 1

Reviewer 1 Report

The review is well-written and summarizes pre-clinical models used for prostate cancer.

1. As there is a strong relation between prostate inflammation and PCa developements, it could be worth to add a paragraph about PCa inflammation models and their pre-clinical utility.

2. To facilitate reading, a table that summarizes various prostate cancer cell lines and their application in pre-clinical studies would be useful.

1. Please check for spelling mistakes as: Promoters instead of promotors.

Author Response

Point-by-point letter for Reviewer 1

I thank the reviewers for the constructive and positive revision. We have applied the majority of the suggestions to the review and find it has improved the manuscript overall. Note, a new table 1 has been added listing cells used for in vivo models of PCa.

The review is well-written and summarizes pre-clinical models used for prostate cancer.

  1. As there is a strong relation between prostate inflammation and PCa developements, it could be worth to add a paragraph about PCa inflammation models and their pre-clinical utility.

We concur with the reviewer's perspective that this is a crucial aspect of PCa. As a response, we have integrated a paragraph into the manuscript, specifically at lines 207-214.

  1. To facilitate reading, a table that summarizes various prostate cancer cell lines and their application in pre-clinical studies would be useful.

We have taken into consideration the recommendations of both reviewer 1 and 3, and as a result, we have incorporated a table listing the commonly used cell lines for xenograft studies.

  1. Please check for spelling mistakes as: Promoters instead of promotors.

We appreciate the reviewer for identifying these spelling mistakes, which have now been rectified.

Reviewer 2 Report

This review article entitled "Pre-clinical models to study human prostate cancer" by Thomsen MK and Busk M indicated preclinical models of PCa for understanding of molecular alterations of propagation in the disorder. This is very important in this field. But, some corrections may be needed. What is the aim and the methodology of the review? What does this review add to the already published papers about the topic? How did you select the info in the review? Please address this matter. The authors should dedicate a section where they explicitly discuss why and how this review article is important for the field. Please list ten keywords chosen from Medical Subject Headings (MeSH). graphical abstract that will visually summarize the main findings of the manuscript is highly recommended. In conclusions, the authors should make an effort to explain the theoretical implications as well as the translational application of their research. I believe that it would be necessary to discuss theoretical and methodological avenues in need of refinement as well as suggestions for a path forward in understanding the importance of this study.

Moderate editing of English language required

Author Response

Point-by-point letter for reviewer 2

I thank the reviewers for the constructive and positive revision. We have applied the majority of the suggestions to the review and find it has improved the manuscript overall. Note, a new table 1 has been added listing cells used for in vivo models of PCa.

Comments and Suggestions for Authors

This review article entitled "Pre-clinical models to study human prostate cancer" by Thomsen MK and Busk M indicated preclinical models of PCa for understanding of molecular alterations of propagation in the disorder. This is very important in this field. But, some corrections may be needed. What is the aim and the methodology of the review? What does this review add to the already published papers about the topic? How did you select the info in the review? Please address this matter. The authors should dedicate a section where they explicitly discuss why and how this review article is important for the field. Please list ten keywords chosen from Medical Subject Headings (MeSH). A graphical abstract that will visually summarize the main findings of the manuscript is highly recommended. In conclusions, the authors should make an effort to explain the theoretical implications as well as the translational application of their research. I believe that it would be necessary to discuss theoretical and methodological avenues in need of refinement as well as suggestions for a path forward in understanding the importance of this study.

Following the reviewer's suggestion, we have included 10 keywords generated by MeSH. As this is a review, we haven't included a graphical abstract. However, Figure 2 could potentially serve as a graphical abstract. We will discuss this with the editor.

Regarding the methodology, this review is not structured as a systematic review but is based on the authors' insights into pre-clinical models for prostate cancer. We have highlighted key work that has contributed to the advancement of the field and have discussed the pros and cons of different models. This distinction is evident in the text and has also been noted by other reviewers. Therefore, we have not specifically outlined an overall goal for the review, as we believe this is addressed in the abstract. We hope that the reviewer concurs with our interpretation of the manuscript.

Reviewer 3 Report

This review written by Thomsen et al describes the pre-clinical models for prostate cancer research, covering many of the advances and some of the pitfalls and future perspectives. The generation of cancer models is of great importance to bring forward cancer research and treatment.

Thus, I recognize this review to be an important and necessary revision of the field's current state. The following suggestions can help to improve the scientific contributions of this review:

 In section 2.3, targeting multiple genes in the prostate; Authors describe the need to combine, for example, Pten, Trp53, and or Rb1 to promote cancer onset and progression. Please refer to how these tumours are relevant and recapitulate human tumours.

 Given the focus of the review it suggested to authors to make a table to list, for example, the genetically engineered mouse models (GEMM) of prostate cancer, model type, engineered genes, strain, features, advantages, and disadvantages.

Similar suggestion of a summary table would be to address the characteristics and other cell lines commonly used in xenograft models.

 Is also suggested to mention to 3D in vitro models of patient-derived prostate cancer xenograft and organoids are alternatives and current disadvantages.

Author Response

Point-by-point letter for reviewer 3

I thank the reviewers for the constructive and positive revision. We have applied the majority of the suggestions to the review and find it has improved the manuscript overall. Note, a new table 1 has been added listing cells used for in vivo models of PCa.

This review written by Thomsen et al describes the pre-clinical models for prostate cancer research, covering many of the advances and some of the pitfalls and future perspectives. The generation of cancer models is of great importance to bring forward cancer research and treatment.

Thus, I recognize this review to be an important and necessary revision of the field's current state. The following suggestions can help to improve the scientific contributions of this review:

In section 2.3, targeting multiple genes in the prostate; Authors describe the need to combine, for example, Pten, Trp53, and or Rb1 to promote cancer onset and progression. Please refer to how these tumours are relevant and recapitulate human tumours.

We have introduced the human relevance of mutations in these common genes in Section 2.3, specifically at lines 134-135.

Given the focus of the review it suggested to authors to make a table to list, for example, the genetically engineered mouse models (GEMM) of prostate cancer, model type, engineered genes, strain, features, advantages, and disadvantages.

This is a valuable suggestion, and we have taken it into account. However, a comprehensive table has already been published by Cory Abate-Shen (PMID: 29661807). To address this, we have included a reference at line 95 (ref 21) and instead presented the essential mouse models in Figure 1.

Similar suggestion of a summary table would be to address the characteristics and other cell lines commonly used in xenograft models.

We have taken into consideration the recommendations of both reviewer 1 and 3, and as a result, we have incorporated a table listing the commonly used cell lines for xenograft studies.

Is also suggested to mention to 3D in vitro models of patient-derived prostate cancer xenograft and organoids are alternatives and current disadvantages.

We have briefly mentioned 3D models as an alternative, recognizing the significance of considering this option alongside in vivo models (line 337-342). We appreciate the reviewer for providing this suggestion.

Reviewer 4 Report

This review wrote by Thomsen et al reviewed the development of preclinical models for studying prostate cancer and discussed the pros and cons of different types of mouse models. This review is well-organized and informative. I suggest it publish on Cancers after a few points revised:

1. The review need a thoroughly check on the citation, some places are lack of references, such as:

“mice with broad genetic alterations often develop cancer in organs other than the prostate”;

“depletion 118 of many different genes has shown…often occurring at a late stage ”;

“metastasis are important features of PCa…with metastatic PCa have a poor prognosis”.

2. “In the 1990s, Greenberg and Zhang utilized…when four elements were used.”:

Describe the reason why Probasin promoter owns the specificity of driving tissue-specific expression in mouse prostate. The same amendment should be made as well for PSA-Cre and Nkx3.1.

3. Figure 1 is incomplete, suggest adding figures to depict the molecular mechanisms of Pten involving in PCa initiation and progression, and draw the pathway as well and placed in Fig. 1.

4. add the full name of “AR”.

5. “Overall, orthotopic delivery has many benefits, and future preclinical models of PCa will rely more on this methodology”: Is there evidence/data to support this opinion? Give your reasons to why future preclinical models of PCa will reply on orthotopic delivery more than other methodology introduced in this review.

6. “As an example, the development and testing of the radiosensitizer Nimorazole…was based exclusively on mouse studies using a single SC tumor model”: why this example can support the opinion of "maintaining the similarity to the average patient tumor is not crucial or even desirable"? Is there any study performed to prove the effectiveness of the SC tumor model is equal/over that of the orthotopic model as pre-clinical model for the development of new therapies for head and neck cancer?

7. Suggest expanding the perspectives, particularly the future opportunities and challenges.

Author Response

Point-by-point letter for reviewer 4

I thank the reviewers for the constructive and positive revision. We have applied the majority of the suggestions to the review and find it has improved the manuscript overall. Note, a new table 1 has been added listing cells used for in vivo models of PCa.

This review wrote by Thomsen et al reviewed the development of preclinical models for studying prostate cancer and discussed the pros and cons of different types of mouse models. This review is well-organized and informative. I suggest it publish on Cancers after a few points revised:

  1. The review need a thoroughly check on the citation, some places are lack of references, such as:

“mice with broad genetic alterations often develop cancer in organs other than the prostate”;

“depletion of many different genes has shown…often occurring at a late stage ”;

We have included references for mice with the loss of either Trp53 or Pten broadly, as these mice develop cancer in other organs as well. See line 124, ref 27-30.

“metastasis are important features of PCa…with metastatic PCa have a poor prognosis”.

We have included the SEER database as a reference.

  1. “In the 1990s, Greenberg and Zhang utilized…when four elements were used.”:

Describe the reason why Probasin promoter owns the specificity of driving tissue-specific expression in mouse prostate. The same amendment should be made as well for PSA-Cre and Nkx3.1.

We have incorporated into the review that these cre lines are primarily regulated by AR (see line 66 and 72).

  1. Figure 1 is incomplete, suggest adding figures to depict the molecular mechanisms of Pten involving in PCa initiation and progression, and draw the pathway as well and placed in Fig. 1.

The Pten pathway has been described from line 103 to 106. We believe that including this in Figure 1 would shift the focus away from the genetic alterations necessary to induce PIN / Cancer / Metastatic disease. Consequently, we have chosen not to follow the reviewer's recommendation.

  1. add the full name of “AR”.

We appreciate the reviewer for bringing this mistake to our attention. We have now addressed it and included the correction at line 66.

  1. “Overall, orthotopic delivery has many benefits, and future preclinical models of PCa will rely more on this methodology”: Is there evidence/data to support this opinion? Give your reasons to why future preclinical models of PCa will reply on orthotopic delivery more than other methodology introduced in this review.

We have outlined some advancements associated with orthotopic delivery compared to the utilization of Cre lines. Additionally, we conducted a study involving both a Cre line and an adenovirus to induce loss of function in the prostatic tissues. In this study, the use of an adenovirus resulted in tumor growth in a portion of the organ, while the majority of the prostate remained intact. We have revised the sentence to emphasize the benefits of orthotopic delivery instead of a more bolt statement.

Revised sentence: Thus, orthotopic delivery offers an advantage by inducing cancer in a specific subset of cells, enabling clonal expansion while also being temporally aligned with the mouse's age.

  1. “As an example, the development and testing of the radiosensitizer Nimorazole…was based exclusively on mouse studies using a single SC tumor model”: why this example can support the opinion of "maintaining the similarity to the average patient tumor is not crucial or even desirable"? Is there any study performed to prove the effectiveness of the SC tumor model is equal/over that of the orthotopic model as pre-clinical model for the development of new therapies for head and neck cancer?

We have expanded this paragraph to point out different advance for SC or orthotopic model. With current technology, pre-clinical testing of localized treatment like radiotherapy using potentially curative treatment doses, thus allowing tumor control probability studies (which clearly is superior to tumor growth delay studies, when it comes to clinical translation) is only feasible in tumors located in an extremity, such as SC in the foot. This fact is clearly stated in the text. See line 308-316 for details.

  1. Suggest expanding the perspectives, particularly the future opportunities and challenges.

We have considered the suggestion to expand the perspectives, but we have decided to maintain conciseness to avoid repetition and to keep the focus on the main message.

Round 2

Reviewer 2 Report

I recommend this manuscript as for publication.

I recommend this manuscript as for publication.

Reviewer 4 Report

acceptance